# Al-Based Nano-Sized Composite Energetic Materials (Nano-CEMs): Preparation, Characterization, and Performance

**DOI:** 10.3390/nano10061039

**Published:** 2020-05-29

**Authors:** Weiqiang Pang, Xuezhong Fan, Ke Wang, Yimin Chao, Huixiang Xu, Zhao Qin, Fengqi Zhao

**Affiliations:** 1The Third Department, Xi’an Modern Chemistry Research Institute, Xi’an 710065, China; xuezhongfan@126.com (X.F.); zhuzhangmangxiewk@163.com (K.W.); xhx204@163.com (H.X.); 2Science and Technology on Combustion and Explosion Laboratory, Xi’an Modern Chemistry Research Institute, Xi’an 710065, China; qzhao087@hotmail.com (Z.Q.); zhaofqi@163.com (F.Z.); 3School of Chemistry, University of East Anglia, Norwich Research Park, Norwich NR4 7TJ, UK; y.chao@uea.ac.uk

**Keywords:** composite energetic materials, nano-sized particles, Al-based, morphology performance, thermal decomposition, hazardous properties

## Abstract

As one of the new types of functional materials, nano-sized composite energetic materials (nano-CEMs) possess many advantages and broad application prospects in the research field of explosives and propellants. The recent progress in the preparation and performance characterization of Al-based nano-CEMs has been reviewed. The preparation methods and properties of Al-based nano-CEMs are emphatically analyzed. Special emphasis is focused on the improved performances of Al-based nano-CEMs, which are different from those of conventional micro-sized composite energetic materials (micro-CEMs), such as thermal decomposition and hazardous properties. The existing problems and challenges for the future work on Al-based nano-CEMs are discussed.

## 1. Introduction

Nano-sized composite energetic materials (nano-CEMs), owing to their large specific surface area, high surface energy, and high surface activity, can realize the nanometer of energetic materials in the process of preparation, storage, and application [1,2]. Moreover, nano-CEMs can integrate the characteristics of each component, thus the new types of energetic materials with various characteristics can be obtained, which can reduce the size of reactants in composites to nano-scale, and then increase the contact interfacial area effectively between materials, so as to solve the problem of slow transmission speed of micro-sized composite energetic materials (micro-CEMs) owing to large particles (chemical reaction rate, mass transfer, and heat transfer), and obtain energetic materials with energy density and a high energy release rate [3,4]. In addition, the energy density of nano-sized energetic composites prepared by the sol–gel method is two times the monomolecular energetic material [5]. The mass transfer speed between the oxidant and the aluminum powder in the pore is fast and the reaction is sensitive. The burning speed is more than 1000 times compared with that of the similar ordinary oxides. Therefore, nano-CEMs as a new type of energetic material have attracted wide interests in the research community. Al is widely used to increase the energy and raise the flame temperature in rocket propellants because of its high combustion enthalpy, easy availability, low toxicity, and good stability in energetic applications, such as propellants, pyrotechnics, and explosives. Especially, nano-sized Al (nAl) is broadly exploited to improve performance incrementing the burning rate and combustion efficiency of energetic systems, leading to shorter ignition delays and shorter agglomerate burning times with respect to propellants containing μAl [6,7,8,9]. In this paper, the developments and achievements in the preparation and performance of Al-based nano-CEMs are reviewed. The preparation methods and properties of Al-based nano-CEMs are emphatically analyzed, and the existing problems and challenges in the future work are reviewed.

## 2. Preparation Methods of Nano-CEMs

Nano-CEMs, being composed of one or more principal energetic components and some additives or binders, have been widely used in military. The uniform dispersing of energetic materials with nano-sized components can display the surface activities, improving the performance of traditional energetic materials and enhancing their practical applications. There are many methods to prepare the nano-CEMs, such as the solid phase method, liquid phase method, and gas phase method (according to the state of raw materials), as well as physical and chemical methods (according to the prepared techniques) [1,5,10]; while there is a big difference for the preparation methods of nano-CEMs mentioned above, here, the mail methods on their preparation were reviewed.

In the last two decades, nano-sized metal matrix composites have witnessed tremendous growth, and particulate-reinforced nanocomposites have been extensively employed in the propulsion field for their tremendous advantages. As we know, non-homogeneous particle dispersion and poor interface bonding are the main drawbacks of conventional manufacturing techniques. A critical review of nanocomposite manufacturing processes is presented; the distinction between ex situ and in situ processes is discussed in some detail. Moreover, in situ gas/liquid processes are elaborated and their advantages are discussed [11]. It provides the reader with an overview of nanocomposite manufacturing methods along with a clear understanding of advantages and disadvantages.

### 2.1. Sol–Gel Method

The sol–gel method is one of the most important preparation methods for nano-CEMs, so the preparation of Al-based nano-CEMs will focus on this method. On the one hand, this method can control the components accurately, and provide the feasibility for obtaining the nano-sized materials with controllable and uniform density; one the other hand, it allows the components to mix directly at the nano-scale.

The method is mainly used to prepare the metal oxide-based nano-CEMs. In order to make the oxidant or fuel into sol, other components are added to form gel. After the gel was formed, the porous aerogels with low density could be obtained by supercritical fluid extraction, and the xerogels with high density could be obtained under slow evaporation and at a certain pressure. Then, the sol–gel forms the main framework, and fuel and the other components are filled in it. The metal oxide sol can be obtained using hydrolyze metal salts and alkoxy metals. Figure 1 shows the typical process for sol–gel method preparation.
M_a_X_b_ + bH_2_O → M_a_(H_2_O)_b_ + bX(1)
M_a_(H_2_O)_b_ + 2b[proton scavnger] → M_a_O_b_ + 2b[proton scavenger-H](2)
M_a_(OR)_b_ + 5H_2_O → Ma(OH)_b_ + bROH(3)
M_a_(OH)_b_ → M_a_O_1/2b_ + 1/2bH_2_O(4)

Nano-sized composite energetic materials of Al/Fe_2_O_3_, RDX (hexogen, cyclotrimethylenetrinitramine)/RF (resorcinol-formaldehyde), and RDX/SiO_2_ were prepared by this method [13,14,15]. Compared with the nano-sized composite particles by means of the mechanical mill method, the thermal decomposition temperature can be forwarded by addition of nano-sized RDX/RF composite energetic materials by the sol–gel method; the mechanical sensitivity reduced as well, indicating that it has great potential application in micro initiating explosive devices.

### 2.2. Solvent/Nonsolvent Method 

The solvent/nonsolvent method for nano-CEMs’ preparation is based on crystallography. Nano- sized cuprous dichromate particles coated with perchlorate crystal were prepared, and the particle size can be effectively controlled by adjusting the mass ratio of solvent to nonsolvent. The catalytic effect of nano-CEMs prepared by this method on ammonium perchlorate can be greatly improved, and the thermal decomposition reaction temperature with ammonium perchlorate was obviously decreased, thus the intensity of the thermal decomposition reaction was enhanced [16]. Nano-sized Al/AP (ammonium perchlorate) composite energetic materials were prepared with the solvent/nonsolvent method, the second thermal decomposition temperature decreased after addition of prepared nano particles, while the total heat release increased [17]. Nano-sized AP/CC (copper chromite) and AP/Fe_2_O_3_ composite energetic materials were also prepared by the same method, which has good efficacy [18].

### 2.3. High-Energy Ball Milling Method

The high-energy ball milling method, which makes the hard ball impact, grind, and stir the raw material intensively according to the rotation and vibration actions, and makes it into nano-sized particles. Nano-sized Al/B/Fe_2_O_3_, AP/CuO, AP/CC (copper chromite), and Al/RDX composite energetic materials were prepared by this method [19,20]. The formation of nano-sized AP/CC composite particles can increase the contact area between ammonium perchlorate and copper dichromate, improving the dispersion and uniformity of the prepared particles, thus enhancing the catalysis effect of copper dichromate and reducing the thermal decomposition temperature of AP (For nano-sized Al/B/Fe_2_O_3_ composition, see Section 3.2.1).

### 2.4. Spray Drying Method

The spray drying method is the process of atomizing small particles and drying them with solution, suspension, emulsion, sol, and so on. It is the main process for preparing submicron and nano-sized particles in industry, and its equipment is simple, low cost, and highly efficient. HMX-(octogen, cyclotetramethylenetetranitramine), RDX-, and CL-20 (hexanitrohexaazaisowurtzitane)-based nano-sized composite energetic materials were prepared by means of the spray drying method, and the process was optimized [21]. Compared with the original particles, the average particle size of prepared HMX-based composite is 30–100 nm, the activation energy and thermal sensitivity are much lower than that of pure HMX, and the impact sensitivity of sample with NC (nitrocellulose) as the binder has little change. For the RDX-based nano-sized composite, the addition of different binders can greatly reduce the impact sensitivity of RDX. For example, with F_2602_ as the binder, the activation energy of nano-sized CL-20/F_2602_ composite increased, the impact sensitivity decreased significantly, while the thermal sensitivity had little change.

### 2.5. Other Methods

Hydrothermal reactions have been widely recognized in natural and artificial systems. Particularly, the formation, production, alteration, and decomposition of all substances and materials in natural systems are always related to the action of water (aqueous solutions) at higher temperatures under different pressures. Hydrothermal/solvothermal is another significant and proposed method to prepare nano-sized energetic materials. The feature and future of hydrothermal/solvothermal reactions for synthesis/preparation of nano-materials with desired shapes, sizes, and structures were elaborated [22].

The nano-sized Al/Al_2_O_3_(p) composites were fabricated from Al-K_2_ZrF_6_-Na_2_B_4_O_7_ system by sonochemistry in situ reaction. The fabrication mechanisms, including high intensity ultrasonic influence on microstructures and reinforcement particles–aluminum matrix interface, were investigated by X-ray diffraction (XRD), scanning electron microscopy (SEM), and transmission electron microscopy (TEM), and the reaction mechanisms were investigated [23]. The results show that the component of the as-prepared composites is Al_2_O_3_ reinforcement. Al_2_O_3_ particles are uniformly distributed in the aluminum matrix, the morphologies of Al_2_O_3_ particles present in a nearly sphere-shape, the sizes are in the range of 20–100 nm, and the interfaces are net and no interfacial outgrowth is observed. The sonochemistry method was reviewed for use in the fabrication of nanomaterials [24]. 

Freeze-drying is another one of the effective methods to prepare the nano-sized composites. Al/AP nano-sized composites containing nano-sized aluminum powder with low density were prepared by means of the freeze-drying process [25,26]; this method has been used in solid rocket propellants. The reduction method, EEW (electro-exploding wire), CVD (chemical vapor deposition), and PCD (photo catalytic degradation) are well known methods to prepare nano-sized energetic materials. Different from the traditional mixture of reductant and oxidant, the porous metal matrix composite has nano-sized holes and an extremely high specific surface area, and the contact area between oxidant and reductant is large, which displays different performances of explosion characteristics [27,28,29,30].

## 3. Characteristics and Performance of Nano-CEMs

### 3.1. Al-Based Binary System

#### 3.1.1. Nano-Sized Al/Fe_2_O_3_ Energetic Composite

Nano-sized Al(s)/Fe_2_O_3_ composites are readily produced from a solution of Fe(III) salt by adding an organic epoxide and metal powder, and nano-CEMs formed with Fe_2_O_3_ as metal oxide and Al as fuel, that is, super thermite, have an extremely fast chemical reaction and the burning rate can reach up to 1000 times that of ordinary thermite, which can thus generate an extremely fast reaction wave and release huge energies [4,10,12]. Nano-sized Al/Fe_2_O_3_ composites were prepared by Gash et al., and the gel structure was observed by high resolution transmission electron microscopy (HRTEM) [21]. It was found that the nano-sized Al/Fe_2_O_3_ energetic composites were composed of 3–10 nm Fe_2_O_3_ clusters that are in intimate contact with ultrafine Al particles with 25 nm in diameter, and the Al particles have an oxide coating with thickness of ∼5 nm. This value agrees well with the pristine ultrafine grain Al powder, indicating that the sol–gel synthetic method and processing does not significantly perturb the metal fuel. Nonetheless, both qualitative and quantitative characterizations have shown that these materials are indeed energetics, while the materials described here are relatively insensitive to standard impact, electrostatic spark, and friction tests. Qualitatively, it does appear that these nano-sized energetic composites burn faster, which are more sensitive to thermal ignition than their conventional counterparts, and the aerogel materials are more sensitive to ignition than those of xerogels.

Nano-sized Al/Fe_2_O_3_ energetic composite was prepared by means of the sol–gel method as well as with the supercritical fluid drying method in China [26,27,31,32]; SEM photographs are listed in Figure 2. The experimental results showed the structure of nano-sized Al/Fe_2_O_3_ composite energetic material was a significant fiber network. The average particle size of Al in the Al/Fe_2_O_3_ composite was 40 nm. The specific surface area of the Al/Fe_2_O_3_ composite was 147.9 m^2^/g, which is much smaller than that of blank Fe_2_O_3_ aerogel. Its average pore size was 8 nm and the distribution of pore size was more uniform. From the results, it was believed that the sol–gel method can provide processing advantages over the conventional methods in the areas of cost, purity, homogeneity, and safety at the very least, and potentially act as energetic materials with interesting and special properties.

#### 3.1.2. Nano-Sized Al/NC Energetic Composite

In order to utilize the full potentials of the existing energetic materials, nano-sized Al/nitrocellulose (Al/NC) composite materials were prepared with the mass ratio of nAl to NC of 0:10, 1:10, 3:10, 5:10, 7:10, and 9:10, respectively, through the sol–gel and supercritical CO_2_ drying methods. The structures of nano-CEMs were characterized by SEM and DSC methods. The physical mixed Al/NC and NC aerogel/Al mixture with the same mass ratio as that of nano-sized Al/NC composites was prepared as the compared composition at the same time. SEM images of different composite samples are shown in Figure 3. Al elements distribution in nano-sized Al/NC energetic composite by EDS (energy dispersive spectroscopy) analysis are shown in Figure 4. The experimental results show that the nano-sized Al/NC composite materials belong to mesoporous material and nAl powder is well-distributed in the gel matrix. The specific surface area of the Al/NC particle decreases with the increasing Al powder mass fraction. When the mass ratio of Al/NC is 5:10, the decomposition heat per unit mass of NC increases from 1689.21 J·g^−1^ to 2408.07 J·g^−1^.

#### 3.1.3. Nano-Sized Al/MnO_2_ Energetic Composite

Owing to the close contact area and uniform distribution between metal fuel and oxidant, nano-structured energetic materials have better energy release performance and ignition performance. Nano-structured energetic composite composed of MnO_2_ nano sheets with root-embedded Al on silicon wafer was prepared [34]. Figure 5 shows the SEM photos of MnO_2_ and nano-sized Al/MnO_2_ composite particles. As shown, the thickness of MnO_2_ grown on the silicon substrate is 900 nm, and the thickness of each sheet is less than 10 nm. The uniform distribution of MnO_2_ nano-flakes on the whole silicon substrate indicates that this method can realize the growth of MnO_2_ on the silicon wafer with a large surface area (Figure 5a,c). Moreover, Al can deposit on the porous MnO_2_ nano sheet successfully, and there was some Al powder at the interface of Si–MnO_2_, indicating that Al could fill in the porous structure of MnO_2_ nano sheet and come into close contact with the MnO_2_ nano sheet (Figure 5b). Additionally, the deposited Al is “rooted” in the porous MnO_2_ and “grows” into a rod-shaped structure, while the surplus rod-shaped Al aggregates and forms a densed vertical “coral” structure (Figure 5d).

Additionally, Al powder with 500 nm, 1000 nm, and 1500 nm thickness was deposited on MnO_2_ with 900 nm thickness, as samples S900-500, S900-1000, and S900-1500 were conducted (Figure 6). Thermal analysis shows that the nano-structured Al/MnO_2_ energetic materials have a lower onset reaction temperature (510 °C) and relatively higher heat of reaction (2380 J·g^−1^) before Al melting. Hydrophobicity is achieved by coating a layer of fluorocarbon, indicating the possible long-term storage ability.

In order to improve the long-term storage stability, fluorocarbon compound was deposited on the surface of nano-sized Al/MnO_2_ composite, which have nano texture structures. Figure 7 shows the SEM photos of Al/MnO_2_/fluorocarbon composite. It can be seen that the structure of Al/MnO_2_ remained unchanged after the fluorocarbon coating, and the surface of the coated composite is better than that of coating before.

#### 3.1.4. Nano-Sized Al/CuO Energetic Composite

CuO is a good catalyst on the decomposition and combustion of solid propellant, as we know. In order to significantly enhance the energetic characteristics and long-term storage stability, two types of super-hydrophobic nano-sized Al/CuO core/shell structured particles were prepared onto Cu foils, with CuO nano-rods or nano-tubes as the core and Al as the shell, combined with the solution chemistry method, magnetron sputtering, and surface treatments. Both nano-sized materials exhibit excellent thermal behaviors, especially for the hollow tube structured Al/CuO nano-tubes. Moreover, the functionalized nano-sized Al/CuO particles possess long-term storage stability for as high as 88% of energy left after being exposed in the air for one month, benefitting from the enhanced resistibility to the humid environment. Combustion analysis of functionalized Al/CuO nanotubes is performed and the results reveal that the steady combustion process with the flame propagation speed is 100 m·s^−1^. This study can provide new ideas for maintaining the activity of nano-CEMs through surface functionalization and significativity for practical applications [35].

A method is disclosed for producing an energetic metastable nano-composite material. Under pre-selected milling conditions, a mixture of powdered components is reactively milled. These components will spontaneously react at a known duration of the pre-selected milling conditions [36]. The milled powder is recovered as a highly reactive nanostructured composite for subsequent use by controllably initiating destabilization thereof. On the basis of this method, several thermite powders with a nominal composition of 2Al·3CuO were prepared by arrested reactive milling using different milling times [37]. The powder particles comprise a fully dense Al matrix with CuO inclusions. The dimensions of the CuO inclusions are around 100 nm and are effectively the same for all prepared powders. The number of inclusions per unit of mass of the composite particles increases with the increased milling times, while unattached CuO particles are present in the samples prepared with shorter milling times. The redox reaction kinetics in the Al/CuO thermites was described, and the multistep reaction model remains valid for composite powders prepared with different milling conditions, and thus characterized by different aluminum/copper oxide ratios. One point should be note that, to accurately describe the rate of redox reactions in specific nanocomposite powders, the model must account for the size of CuO inclusions and the number of inclusions per unit of mass in the nanocomposite particle.

#### 3.1.5. Nano-Sized Al/Co_3_O_4_ Composite

The energetic performance of nano-CEMs depends on the interfacial diffusion and mass transfer during the reacted process. However, the development of the desired structure to significantly enhance the reactivity still remains challenging for researchers. Co_3_O_4_ micro-spheres with 3D porous hollow were designed and prepared, in which gas-blowing agents (air) and maximized interfacial interactions were introduced to enhance the mass transport and reduce the diffusion distance between the oxidizer and fuel (aluminum) [38]. A low-onset decomposition temperature (423 °C) and high heat output (3118 J·g^−1^) were observed, resulting from porous and hollow nano-structure of Co_3_O_4_ micro-spheres. Furthermore, 3D hierarchical Al/Co_3_O_4_ arrays were directly fabricated on the silicon substrate, which was fully compatible with silicon-based microelectromechanical systems to achieve functional nano-sized energetics-on-a-chip. This approach can provide a simple and efficient way to fabricate the 3D ordered nano-sized energetic arrays with superior reactivity, and it has potential applications in micro-energetic devices.

#### 3.1.6. Nano-Sized Al/MoO_3_ Energetic Composite

A monolayer of two-dimensional (2D) MoO_3_ with 3–4 μm thickness was prepared by the ultrasonic dispersion method, and then the nano-sized Al/MoO_3_ composite with a high interface contact area by combining Al nanoparticles with 80 nm in diameter was prepared [39]. The combustion data show that the burning rate reaches (1730 ± 98.1) m·s^−1^ for their high interface contact area, the increasing rate of pressure is (3.49 ± 0.31) MPa·s^−1^, and the temperature in the reacted area is 3253 K [40]. This is the highest value for the nano-sized Al/MoO_3_ CEMs reported so far. At the same time, the Al/Fe_2_O_3_ and other metal oxides-based energetic materials were coated on Al/Monel (Monel nickel-based alloy), while Ni/Al, Ni/Si, and other laminated structural materials were prepared as well using the sol–gel method and multi-layer injection method; see details in [41].

Moreover, low-temperature redox reactions in nano-thermites prepared by arrested reactive milling (ARM) were described by the Cabrera–Mott (CM) mechanism, in which the growth of very thin oxide layers is accelerated by an electric field induced across such layers. A reaction mechanism combining the initial CM step with the following oxidation steps identified earlier for oxidation of Al and representing growth and phase transitions in various polymorphs of alumina was proposed and shown to be valid for different Al/CuO nanocomposite powders prepared by ARM. Al/MoO_3_ nanocomposite was prepared by the similar multi-step reaction mechanism [42]. The powder particles comprise a fully dense Al matrix with nano-sized MoO_3_ inclusions. A weight loss representing dehydration of the composite material was observed at low-temperature thermogravimetry (TG) measurements. Kinetics of the initial exothermic redox reaction observed in both DSC and micro-calorimetry experiments could be successfully described by the CM mechanism. Reactions at elevated temperatures were also well described by the previously identified sequence of steps associated with phase changes and oxidation of various alumina polymorphs. A modification to the CM kinetics was necessary to describe an intermediate range of temperatures, between approximately 400 and 600 K. This modification is suggested to represent a temperature induced change in the structure of a very thin precursor aluminum oxide layer separating Al and MoO_3_.

#### 3.1.7. Nano-Sized Al/Ni Energetic Composite

As we know, under the stimulation of external conditions (such as heating, current, and laser), Al/Ni composite can produce solid–solid exothermic reaction with a high temperature, and maintain the self-sustaining reacted process, forming the compounds between these two elements. Further, it has been used in the welding and ignition research fields [43]. On the basis of the high temperature and exothermic reaction of Al/Ni composite with a large specific surface area of nano-sized particles, a new nano-structured Al/Ni energetic composite on silicon (Si) substrate using the template method were prepared [44]. Figure 8 shows the SEM photo and DSC curve of the Al/Ni nano-sized composite. As can be seen, Ni nano-sized rods (50 nm in diameter) grow vertically on the substrate, and Al covers the surface of Ni nano-sized rods and is partially embedded into the array of Ni nano-sized rods to form an embedded nano-sized structure (Figure 8a). This kind of embedded nano-sized structure can increase the contact area between the reactants (Al and Ni) and increase their solid–solid reaction rate. The ignition of nano-sized Al/Ni composite was examined by a single pulsed laser irradiation technique, and the results show that the nano-sized Al/Ni composite produces sparks for several milliseconds, which is advantageous to its explosion application. The 50% firing energy of the Al/Ni energetic composite is 36.28 mJ, with an energy density of 46.22 mJ·mm^−2^.

The core–shell shaped carbon coated nano-sized aluminum (nAl) in the atmosphere of CH_4_ and Ar gases by means of laser induction combined heating method was prepared [5]. It was shown that its core is crystalline aluminum and its shell is graphite like carbon. Most of the particles are spherical with a diameter of 20–60 nm (thickness is 3–8 nm). During the formation of core–shell shaped nano-sized Al/C particles, owing to the low density of carbon and its poor solubility with aluminum, carbon atoms do not dissolve in the interior of the nanoparticles, but they are deposited on the surface of the nanoparticles.

Additionally, nano-sized Al–Ni energetic material with 90–100 nm particles is fabricated and the experimental results of its properties are presented [45]. High values of specific energy and the rate of its release make it possible to use this material as a heat release element in thermoelectric power generation devices. It has been demonstrated experimentally that it is possible to maintain a voltage value higher than 1 V for 45 s as a result of combustion of a 3 g Al–Ni sample, and that using a simple DC–DC converter will allow charging supercapacitors or accumulators.

#### 3.1.8. Nano-Sized Al/Fatty Acid Energetic Composite

Aluminum-oleic acid composite nanoparticles with a mean diameter of 85 nm were successfully prepared by means of a wet chemical process [46]. The Al/oleic acid molar ratio affects the thickness of the oleic acid layer on Al nanoparticles. Al electrodes can be formed by firing an Al nanoparticle paste film at 600 °C, and the firing temperature is about 300 °C lower than that required for micrometer-sized Al particles. The electrode formed from commercial Al nanoaparticles is not electrically conducted because of the oxide layer Al nanoparticles; however, the film from the prepared Al nanoparticles for an Al/oleic acid molar ratio of 1:0.05 has a minimum specific resistance of 5.6 mΩ·cm.

#### 3.1.9. Nano-Sized Al/Polymer Energetic Composite

Polyimide (PI) composites with high dielectric permittivity have received a great deal of attention in embedded capacitors and energy-storage devices owing to its excellent thermal stability and good mechanical properties. Nano-aluminum (Al) particles were introduced into PI to prepare promising PI/nano-Al composites [47]. The results indicate that the dielectric constant of the composite films increased with the increase of nano-Al contents, and the highest dielectric constant was 15.7 for a composite film incorporating 15 wt % nano-Al. The effects of mixture doping concentration on volume resistivity and loss tangent are analyzed. The correlation effects of the Al nanoparticles on the different factors that influence the dielectric performance in the PI matrix such as microstructure, resistivity, and interface of the composites were discussed in detail. This composite film would thus possess potential application in flexible energy-storage devices.

### 3.2. Al-Based Ternary System

#### 3.2.1. Nano-Sized Al/B/Fe_2_O_3_ Energetic Composite

In order to reduce the acidic impurity of B_2_O_3_ and H_3_BO_3_ on the surface of amorphous boron powder and improve the efficiency of boron powder application, the nano-sized Al/B/Fe_2_O_3_ energetic composites were prepared through the high energy ball milling [48]. It was found in Figure 9 that the particle size of raw Al and B is micro-scale; with the increase of milling time, the average particle size of Al/B bimetal composite decreases. This is because of the large plastic deformation of the two mixed powders in the composite at the initial stage of ball milling. In the physical mixed samples, the spherical particles of aluminum can still be seen. Boron powder and iron oxide are simply adsorbed onto the surface of aluminum powder, indicating that aluminum powder, iron oxide, and boron powder are simply blended, and the morphology of the grains has not changed. For the Al/B/Fe_2_O_3_ tri-metal composite, there are no spherical aluminum particles, flake boron powder, and flocculent iron oxide, which shows that these three components have been combined and the average particle size is about 90 nm. The heat release of the prepared nano-sized Al/B/Fe_2_O_3_ composite increases with an increase in the mass fraction of B in the composition, as can be seen from the DSC curves in Figure 10.

#### 3.2.2. Nano-Sized Al/CuO/KClO_4_ Energetic Composite

A solvent/non-solvent synthetic approach was utilized in preparing nano-sized Al/CuO/KClO_4_ composite by coating Al/CuO particles with a layer of nanoscale oxidizer KClO_4_. The results reveal that, after ball milling and the chemical synthesis process, the phase compositions were not changed. Scanning electron microscopy (SEM) images show that these energetic nano-sized composites consist of small clusters of Al/CuO that are in intimate contact with a continuous and clear-cut KClO_4_ layer (100–400 nm). High K/Cl intensity on the perimeter of the nano-sized particles and high Al/Cu mass fraction in the interior powerfully demonstrated the Al/CuO/KClO_4_ core–shell nanostructure. Electrical ignition experiments and pressure cell test prove that these nano-sized energetic composites are more sensitive to ignition with a much higher burning rate than that of traditional formulations (conventional counterparts). The TG and DSC results show that the burning rate of these energetic nano-sized composites nearly tripled [49].

#### 3.2.3. Nano-Sized Al/B/Ni Energetic Composite

In order to improve the dispersion of nano-sized catalyst particles in the propellant composition and make full use of its catalytic performance, nano-sized Al/B/Ni composites were prepared [50]. It can be seen that the surface of raw Al is smooth, and their average particle size is 20–30 nm (Figure 11a). The coated particles on the surface of nano-sized composite particles are uniform and continuous, and the smooth surface of core aluminum particles is not exposed, thus the coating is relatively complete, and most of their average particle sizes are 20–30 nm, while a few are 50 nm (Figure 11b). Moreover, the proportion of Ni/B alloy particles in the composite particles in the presence of original unit particles increased significantly, and the proportion of agglomerated particles decreased significantly, indicating that the preparation of nano-sized composite particles can improve its dispersion performance in the composition. It can be also found that a thin coating film is attached on the surface of aluminum powder, and the coating layer is uniform and continuous. There are two obscure rings (Figure 11d), which is similar to Figure 11c.

A certain proportion of nano-sized Ni/B amorphous alloy and AP are ground and compounded on a planetary ball mill in the presence of a small amount of anhydrous ethanol, and then dried to obtain the nano-sized Ni/B composite particles and AP. Figure 12 shows the DTA (differential thermal analysis) curves of AP with different mass ratios of nano-sized Al/B/Ni composite particles, and the mass fraction data refers to the percentage of Ni/B in Ni/B/Al and AP composites, which is 5%, 10%, and 12%, respectively. It is shown that, with the increase of the mass fraction of nano-sized Ni/B particles, the exothermic peak temperature of AP decreases, indicating that the catalytic effect increased with an increase in the mass fraction of the nano-sized composite. However, when the mass fraction is more than 10%, the exothermic peak temperature cannot continue to decline, and the decomposition speed decreases, indicating that the catalytic effect does not continue to increase with its mass fraction. Moreover, when the mass fraction of nano-sized Ni/B is 12%, the exothermic peak temperature of composites is higher than that of pure AP at low-temperature range, indicating that nano-sized Ni/B particle hinders the low-temperature thermal decomposition of AP. Additionally, the minimum (or maximum) value of the quadratic curve equation is used to get the optimum amount of catalyst in the graph, as well as the fitting equation between the decomposition temperature and mass fraction of each sample in Figure 12, which is *y* = 1.363*x*^2^ − 21.625*x* + 475.77, where the fitting coefficient is 0.97, it was calculated that the optimum mass fraction of Ni/B particles in Al/NiB composite is 7.93%, and the corresponding minimum temperature is 389.98 °C on the theoretical high-temperature decomposition, indicating that the catalytic effect of nano-sized particles has been improved significantly after treatment.

#### 3.2.4. Nano-Sized Al/RDX/Fe_2_O_3_ Energetic Composite

The sol–gel method is one of the most important techniques for preparation of nano-sized particles, while nano-sized Al/RDX/Fe_2_O_3_ composites with a mass fraction of 85% hexogen (RDX), 3.75% aluminum, and 11.25% iron oxide (Fe_2_O_3_) were prepared by sol–gel template and supercritical CO_2_ fluid drying technology, where mechanical sensitivity and detonation velocity were compared with those of pure RDX. Particles were characterized by scanning electron micro-scale (SEM) shown in Figure 13. Fe_2_O_3_ aerogel is a complete “honeycomb”, which has a relatively uniform pore size, and the average pore size is between 50 nm to 100 nm. When the initial particles are polymerized to form gel, the particles are combined to form a three-dimensional porous aerogel skeleton. After supercritical fluid drying, the liquid in the pores is taken away, forming the honeycomb structure of Fe_2_O_3_ (Figure 13a). A small amount of RDX grains are embedded on the surface of the crystal, and they are formed with the evaporation and diffusion of dimethylformamide (DMF) during the drying process. The structure of Fe_2_O_3_ colloid is broken and a small amount of RDX grains are exposed (Figure 13b). The nano-sized composites in high-resolution microscopic SEM photos exhibit a spherical or near-spherical shape, and the average particle size is in the range of 100–200 nm (Figure 13c).

For this formation mechanism of nano-sized composite, DMF solution of the explosive is filled inside the gel mesh in the process of forming Fe_2_O_3_ gel, and the nAl powder is evenly dispersed in the gel system by ultrasonic vibration instrument. Using the supercritical fluid drying technology by the super solubility and diffusivity of supercritical CO_2_ fluid, RDX rapidly recrystallizes around the nAl powder in the gel hole, which suppressed the further crystallization and growth of RDX effectively. The nano-CEMs were formed from nano-sized RDX and nAl powder encapsulated by Fe_2_O_3_ colloid.

Under the same experimental conditions, the impact sensitivity and friction sensitivity of nano-sized Al/RDX/Fe_2_O_3_ composites are much lower than that of pure RDX (Table 1). This can be related to the “hot spot” formation of the samples during the test process. On the one hand, the grains of nano-sized composites are much smaller, its surface structure integrity is good, and there are few defects, which make it difficult to form local “hot spots” inside. On the other hand, nano-sized particles also have a unique lubrication effect, reducing the friction between the particles effectively, thus the smaller size of the “hot spot” decreases the mechanical sensitivity. The detonation velocity of nano-sized composites is 7185 m·s^−1^, which is higher than that of pure RDX by 615 m·s^−1^. This may be because of the difference between the nano-sized composites and the micro-scale mixed explosives. The mass transfer rate between the explosives or between explosive and metal for micro-scaled explosives is slow, while nano-scaled energetic transfer between explosive particles would not be affected by the mass transfer rate, which contributes to the high detonation velocity [51].

In order to compare the properties with Al/RDX/Fe_2_O_3_, nano-sized B/RDX/Fe_2_O_3_ (mass ratio is 2:90:8) composites were prepared by adding RDX and B powder into the gel template of Fe_2_O_3_, which were prepared by the sol–gel process and dried by supercritical CO_2_ fluid drying technology [52]. The results indicate that the average particle size of nanocomposite energetic material is 30–50 nm. Compared with the pure RDX, the onset thermal decomposition temperature of B/RDX/Fe_2_O_3_ increases by 7 °C, the reaction heat increases by 885 J·g^−1^, and the impact sensitivity (characteristic drop height, *H*_50_) is 40.8 cm.

#### 3.2.5. Nano-Sized Al/RDX/SiO_2_ Energetic Composite

With the continuous improvement of the performance requirements for energetic materials, RDX is one of the most widely used single explosives. In order to reduce the mechanical sensitivity of RDX and improve its thermal decomposition performance, three types of Al/RDX/SiO_2_ with 30%, 50%, and 70% of Al/RDX in the composites were prepared by means of the sol–gel method (Figure 14). In Figure 14a, SiO_2_ shows a cellular network structure. Granular Al/RDX is filled in only a few holes in the SiO_2_ skeleton, owing to the small proportion of Al/RDX (Figure 14b). With the increase of the Al/RDX mass fraction, the filling amount of Al/RDX in the SiO_2_ skeleton increased significantly, and only few holes are not filled in (Figure 14c). All holes in the SiO_2_ skeleton are filled by Al/RDX, which leads to the obvious accumulation of Al/RDX particles in the skeleton (Figure 14d).

On the basis of the preparation of nano-sized Al/RDX/SiO_2_ composite particles, its effects on the thermal decomposition of RDX were investigated by DSC and TG techniques at the heating rate of 10 °C·min^−1^ (Figure 15). The melting and thermal decomposition peak temperatures of RDX in the prepared nano-sized Al/RDX/SiO_2_ composite materials are 1.56–4.49 °C and 18.9–22.4 °C, respectively, which are earlier than those of pure RDX (Figure 15a). With the increase of Al/RDX mass fraction in the SiO_2_ skeleton, the decomposition peak temperature with heat release of RDX increases gradually in the composite. This may be because, with the increase of Al/RDX mass fraction in the composition, the SiO_2_ skeleton is relatively reduced, and the agglomeration and accumulation of fillers occur in the skeleton, which results in the decrease of contact area between the reactants, thus the exothermic decomposition peak temperature of RDX in composites slightly increases on the macro-scale. Moreover, with the increase of Al/RDX mass fraction in nano-sized Al/RDX/SiO_2_ composites, the mass loss of RDX in the composites is gradually delayed (Figure 15b). The reason may be that, with the increasing mass fraction of Al/RDX filler in the SiO_2_ framework, some holes in the SiO_2_ framework are filled unevenly, collapse, and grain agglomeration, which reduces the contact area between the reactants. The mass transfer and heat transfer in the chemical reaction process are adversely affected, which is reflected in the TG curves (Figure 15b), that is, the mass loss of RDX in nano-sized Al/RDX/SiO_2_ composites increased with the increase of the Al/RDX mass fraction.

At the same time, the mechanical sensitivities of nano-sized Al/RDX/SiO_2_ composites were tested, which were compared with the particles by means of mechanical mixed procedure. The friction sensitivity and impact sensitivity of nano-sized Al/RDX/SiO_2_ composites (26–68%, 27.1–95.6 cm) is much lower than those of pure RDX (10%, 134.9 cm) and mechanical mixed ones (74–85%, 18.6–25.1 cm).

In order to improve the combustion of RDX/Al/SiO_2_ composites, nano-sized Al/RDX/AP/SiO_2_ composites were prepared and characterized [54]. As shown in Figure 16a, the morphology of the spherical composite is uniform, the average particle size (higher than 200 nm) and gradation are reasonable, and their accumulation is close. During the dry process, as the solvent evaporates, the gel pores collapse gradually. AP and RDX with solvent diffuse from one gel hole to another gel hole or gel surface (Figure 16b). For the gel complex by the supercritical drying process (Figure 16c), supercritical CO_2_ rapidly penetrates into the gel hole and extracts the organic solvents instantaneously, so that a large number of AP and RDX can crystallize rapidly in each gel hole. Supercritical CO_2_ has super solubility and diffusivity, which can effectively inhibit the diffusion and transfer of RDX between adjacent gel pores, thereby maintaining the integrity of the gel void structure.

Moreover, the mechanical sensitivities of nano-sized Al/RDX/AP/SiO_2_ composites were tested, which were compared with the particles by means of mechanical mixed procedure. The impact sensitivity of nano-sized Al/RDX/SiO_2_ composites (48.3 cm) is much lower than that of the mechanical mixed one (30.7 cm) [55].

In addition, nano-sized Al/AP composites were prepared by means of the freeze-drying process by Martin et al., nAl powder was added to AP solution, and then the mixture solution was quickly poured into the container filled with liquid nitrogen to obtain the low-density nano-sized Al/AP composites, which improve the combustion performance of propellants [56,57].

### 3.3. Properties and Applications of Typical Nano-CEMs

At present, most studies use organic solvent as dispersant to disperse nAl powder and oxide. Considering the high activity of nAl and high flammability in the organic solvent, the preparation process is very dangerous. One alternative method is to use water as the dispersant. Because of the high activity of nAl, the nAl powder can still react with water, resulting in the decrease of activity, although the surface is passivated by a layer of oxide [58,59]. It was found that, based on silane or oleic acid, the organic surface coating can effectively inhibit the reaction between nAl powder and water, ensuring its activity in a long period of time.

The impact ignition energy of nano-sized Al/MoO_3_ composites is 0.3–1.7 J, which increases with an increase in the particle size. The maximum reaction rate is related to the initial particle size, and the reaction rate increases with the decrease of the particle size. The combustion mechanism and reaction properties of four types of nano-sized Al/MoO_3_, Al/WO_3_, Al/CuO, and Al/Bi_2_O_3_ composites were conducted; it was found that the propagation speed depends on the generation of gas and the thermodynamic state of reaction products, and the burning rate decreases with the increase of mixture density, which is contrary to the traditional explosion theory, so the combustion propagation mechanism is different from that of the traditional explosion theory. Nano-sized Al/Cr_2_O_3_ with 10–15 nm in diameter was prepared, and the relationship between particle size and combustion performance by means of CO_2_ laser ignition technique was conducted; the performance is significantly improved compared with that of micro-scaled particles. One interesting point is that nano-sized Al/WO_3_·H_2_O composites were prepared by means of the wet chemical method, and its performance was compared with that of nano-sized Al/WO_3_ composites; it was found that the energy release rate of reaction of Al/WO_3_·H_2_O is significantly improved by the reaction between nAl and H_2_O in WO_3_·H_2_O to produce hydrogen, which increased the total heat release and energy release rate of the reaction.

Three nanocomposite materials with the same nominal stoichiometric thermite composition of Al/CuO were prepared by three different methods: ultrasonic mixing (USM) of constituent nanopowders, electrospraying (ES), and arrested reactive milling (ARM). The combustion performances of the three prepared types of Al/CuO nanocomposites were conducted [60]. Prepared powders were placed in a 6.7 mm diameter, 0.5 mm deep cavity in a brass substrate and ignited by electro-static discharge. The experiments were performed in air, argon, and helium. The mass of powder removed from the sample holder after ignition was measured in each test. Using a multi-anode photo-multiplier tube coupled with a spectrometer, time-resolved light emission traces produced by the ignited samples were recorded in the range of wavelengths of 373–641 nm. Time-resolved temperatures were then determined by fitting the recorded spectra assuming Planck’s black body emission. Temporal pressure generated by ignition events in the enclosed chamber showed that the powder’s combustion properties were tied to both their preparation technique as well as the environment in which they were ignited. The agglomeration of nanoparticles hindered the combustion of USM powders, while it was not observed for the ES powders. Lower temperatures and pressures were observed in oxygen-free gas environments for USM and ES powders prepared using starting nano-particles. For the ES powders, the effect of gas environment was less significant, which was interpreted considering that ES materials included gasifying nitrocellulose binder, enhancing heat, and mass transfer between individual Al and CuO particles. Higher pressures and temperatures were observed in inert environments for the ARM-prepared powder.

Addition of reactive nanocomposite powders can increase the burn rate of aluminum, and thus the overall reaction rate of the energetic formulation. Replacing only a small fraction of the fuel by a nanocomposite material can enhance the reaction rate with little change to the thermodynamic performance of the formulation. The nanocomposite materials Al/3CuO and Al/MoO_3_ prepared by ARM, a scalable “top-down” technique for manufacturing reactive nano-materials, were added to micron-sized aluminum powder and the mixture was aerosolized and burned in a constant volume chamber with varied oxygen, nitrogen, and methane atmosphere [61]. The resulting pressure traces were recorded and processed to compare different types and amounts of modifiers. Additives of nanocomposite powders of Al/MoO_3_ to micron-sized aluminum were found to be effective in increasing both the rate of pressure rise and maximum pressure in the respective constant volume explosion experiments. It was observed that 20 wt % of additive resulted in the best combination of the achieved burn rate and pressure.

The potential application of nano-aluminum/nitrocellulose mesoparticles as an ingredient for solid composite rocket propellants was investigated [62]. The basic strategy is to incorporate nanoaluminum in the form of a micrometer scale particle containing a gas-generator, to enable easier processing and potential benefits resulting from reduced sintering prior to combustion. The mesoparticles were made by electrospray and comprised aluminum nanoparticles (50 nm) and nitrocellulose to form micrometer scale particles. Eighty percent solids loaded composite propellants (AP/HTPB (hydroxilterminated polybutadiene) based) were made with the addition of micrometer sized (2–3 μm) aluminum (10 wt %), and compared directly to propellants made by directly substituting aluminum mesoparticles for traditional micrometer sized particles. The propellant burning rate was relatively insensitive for mesoparticles containing between 5 wt % and 15 wt % nitrocellulose. However, direct comparison between a mesoparticle-based propellant and a propellant containing micrometer scale aluminum particles showed burning rates approximately 35% higher, while having a nearly identical burning rate exponent. High-speed imaging indicates that propellants using mesoparticles have less agglomeration of particles on the propellant surface. 

The results show that the burning rate of nano-CEMs is the fastest when the particle size of Al powder and oxide is at nano-metric level, while the burning rate of CEMs composed of micron Al powder and nano-metric oxide is much faster than that of nanometer Al powder and micro-metric level.

## 4. Conclusions and Future Challenges

As one of new type of functional material, nano-sized composite energetic materials (nano-CEMs) have broad application prospects. However, most of the studies are carried out at the theoretical and laboratory level at present, which is still a long way from engineering and practical application. In the research field of high-energy explosives, the nano-sized composite process can cause the surface of each component of explosive to fully contact each other. On the one hand, it can decrease the agglomeration of ultrafine particles in explosives or improve the physical and chemical properties of explosives. On the other hand, it can also adjust the sensitivity of explosives and improve the safety performance of explosives. In the research field of solid propellants, using the nano-sized composite synthesis technologies, the powder and ultrafine catalyst are prepared into ultrafine composite particles, which can increase the contact area, improve the actual catalytic ability of the catalyst, and then greatly improve the energy release efficiency of the powder [63].

Nano-sized composite energetic materials (nano-CEMs) possess the characteristics of high energy density, high combustion rate of a few kilometers/second, and micron scale critical reaction propagation size. They have shown good application potential in many aspects, such as micro electro mechanical systems (MEMS) devices, anti-infrared decoy materials, and high energy additives. In addition, the combination between oxidants and reducers exhibits unique reaction dynamics characteristics at nano-scale, such as particle size dependence, mass transfer diffusion, energy release, and other mechanisms, which are quite different from the traditional micro-sized scale solid-state reaction. In the past years of study, significant progresses have been made in developing preparation methods, investigating properties, fundamental theory, and applications of nano-CEMs. However, the following difficult challenges to be addressed in the future have been identified.

(1) Understanding of reaction propagation mechanism of nano-CEMs is still in its infancy.

On the basis of the “melting diffusion mechanism” and “metal oxygen turnover mechanism”, only the mass diffusion and reaction rate at the interface were considered, which can simply help us to understand the escape of oxygen atom (O) in oxide at the initial stage (ignition stage) of reaction and the process of combining with active metal. In fact, the reaction propagation mechanism is more complicated after ignition during combustion. In the face of this challenge with complex and multiple factors, an effective strategy is to combine the experiment and theoretical model by peeling off the single influencing factor. However, these models are too simple, and there are few experiments to describe the multiple factors [1,2,64,65,66,67]. Thus, the understanding of the ignition and reaction propagation mechanism of nano-CEMs still requires the joint efforts of colleagues all over the world.

(2) The performance breakthrough is also limited by loading density, stability, and other factors.

Although nano-CEMs have high bulk energy density, it is difficult to obtain the ideal output energy, such as high pressure and shock wave owing to the lack of gas release within the reaction products. In order to obtain a high burning rate, it is necessary to heat the air between the powder voids to accelerate the convection and mass transfer, which significantly increases the dependence of reaction speed and loading density. Although the reaction speed can be slightly increased by the design of hollow spheres, branch, and other microstructures, it is difficult to balance the contradiction between the high bulk density and reaction rate of the composition. The reaction speed of the composition could be slightly improved by introducing gas products through nano-CEMs’ composition design. However, it will bring new problems, such as stability and storage performance [1,2,37,38,39].

(3) Engineering application needs new breakthroughs and deep investigations.

Because most nano-CEMs use highly active nano-sized metals (such as Al, Mg) as reducing agents, how to avoid or reduce the self-oxidation of the highly active metals is the prime problem in engineering applications. For nano-sized aluminum powder, it is easy to agglomerate and influence its dispersion compositions by coating nano Al powder with polymer or other inert materials. In addition, in order to ensure the coating quality, the mass fraction of the inert coating materials is often high, which would reduce the energy density of the composition and increase the complexity of the reaction. In the process of formulation design, it is very difficult to bond the nano-CEMs with high specific surface area using the few mass fractions of binder [1,2,41,42,43,44,45], which is one of the common problems of nano-CEMs in the practical application.

## Figures and Tables

**Figure 1 nanomaterials-10-01039-f001:**
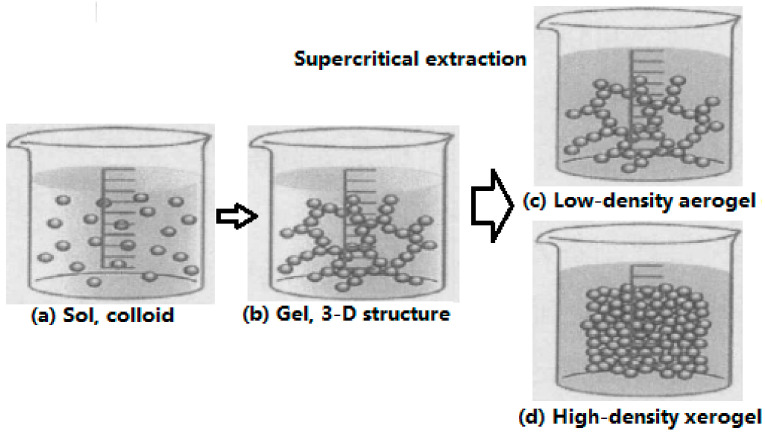
Typical process for sol–gel method preparation. (**a**) Sol, colloid; (**b**) Gel, 3D structure; (**c**) Low-density aerogel; (**d**) High-density xerogel. Reproduced from [12], with permission from Elsevier, 2001.

**Figure 2 nanomaterials-10-01039-f002:**
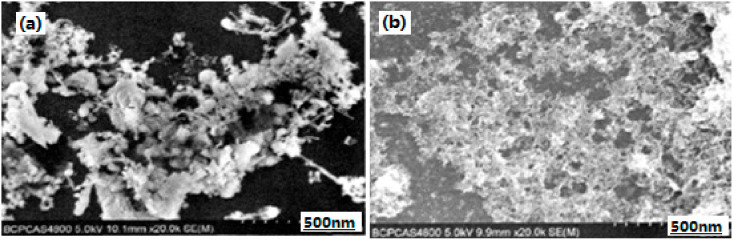
Scanning electron microscopy (SEM) photographs of Fe_2_O_3_ aerogel (**a**) and Al/Fe_2_O_3_ composite energetic material (**b**). Reproduced from [31], with permission from Huo Zha Yao Xue Bao, 2010.

**Figure 3 nanomaterials-10-01039-f003:**
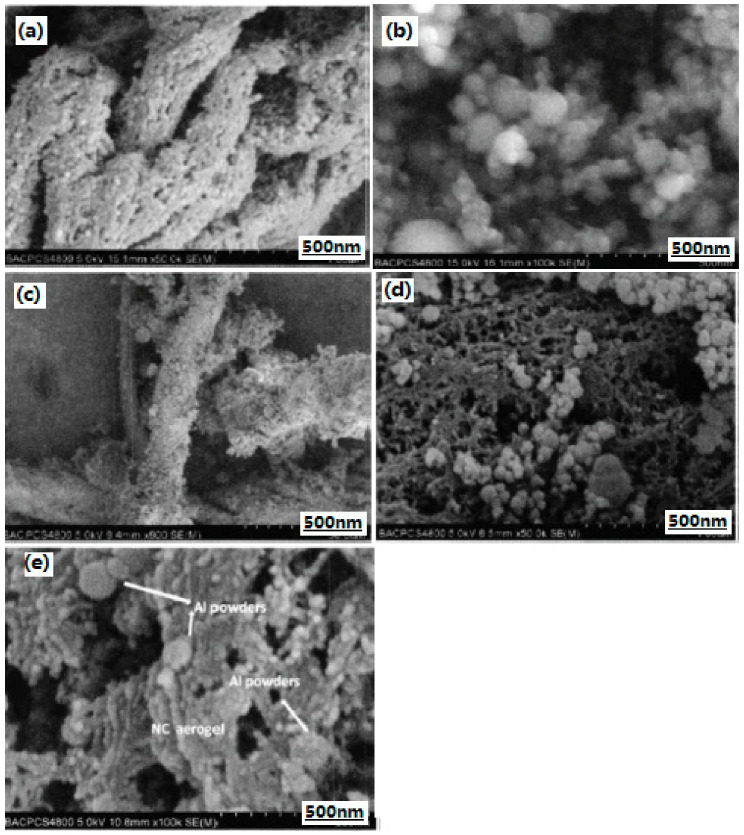
SEM images of different composite samples. (**a**) NC aerogel; (**b**) Al powder; (**c**) Al/NC mixture; (**d**) NC aerogel/Al mixture; and (**e**) Al/NC nano-sized composite. Reproduced from [33], with permission from Han Neng Cai Liao, 2013.

**Figure 4 nanomaterials-10-01039-f004:**
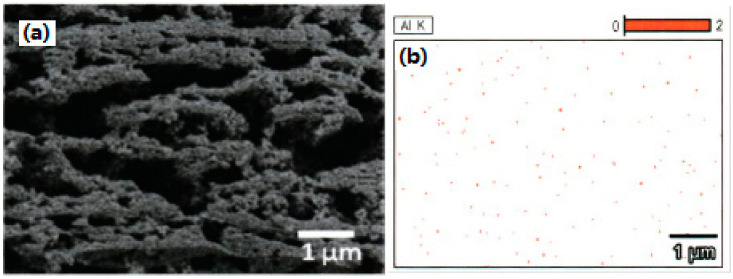
Al elements distribution of Al/NC nano-composite (Al/NC = 1:2) by EDS analysis. (**a**) Selected area of Al/NC nano-sized composite; (**b**) Al elements’ distribution. Reproduced from [33], with permission from Han Neng Cai Liao, 2013.

**Figure 5 nanomaterials-10-01039-f005:**
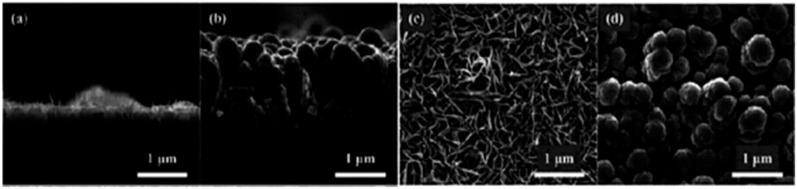
The cross-sectional and top-view SEM photos of MnO_2_ and MnO_2_/Al particles. (**a**) Cross-sectional of MnO_2_; (**b**) cross-sectional of MnO_2_/Al; (**c**) top-view of MnO_2_; and (**d**) top-view of MnO_2_/Al. Reproduced from [34], with permission from Initiators & Pyrotechnics, 2018.

**Figure 6 nanomaterials-10-01039-f006:**
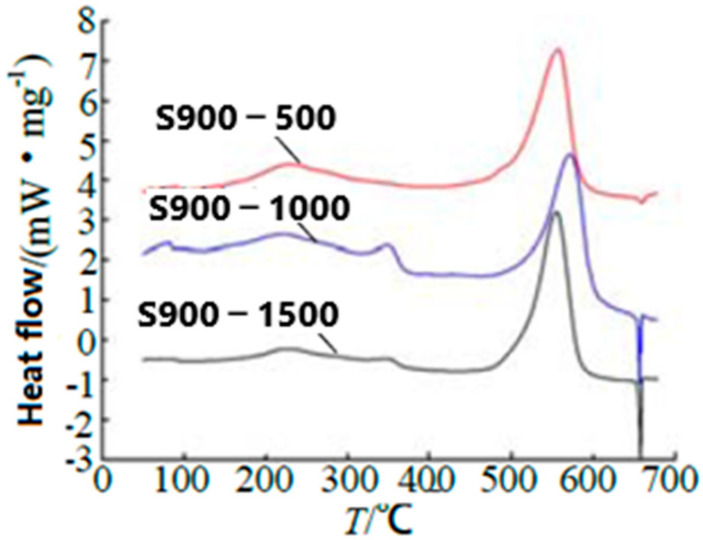
DSC curves of 900 nm thickness MnO_2_ samples with nominal 500 nm, 1000 nm, and 1500 nm Al. Reproduced from [34], with permission from Initiators & Pyrotechnics, 2018.

**Figure 7 nanomaterials-10-01039-f007:**
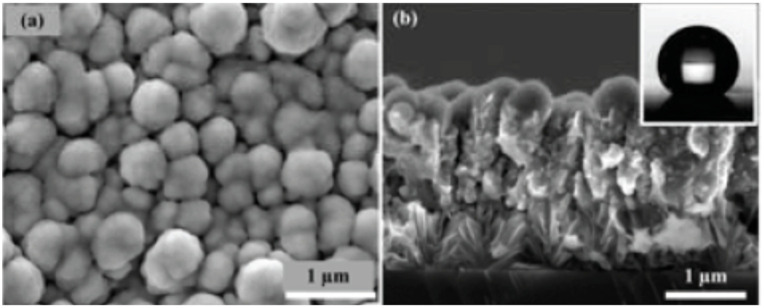
SEM photos of MnO_2_/Al/fluorocarbon composite. (**a**) Cross-sectional; (**b**) top-view. Reproduced from [34], with permission from Initiators & Pyrotechnics, 2018.

**Figure 8 nanomaterials-10-01039-f008:**
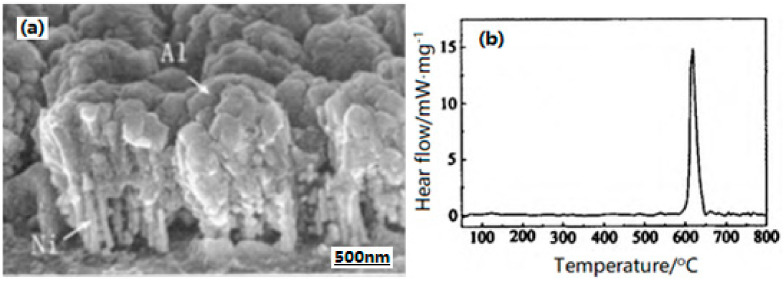
SEM and DSC curve of Al/Ni nano-sized composite. (**a**) SEM of Al/Ni nano-sized composite; (**b**) DSC curve of Al/Ni nano-sized composite. Reproduced from [44], with permission from Huo Zha Yao Xue Bao, 2012.

**Figure 9 nanomaterials-10-01039-f009:**
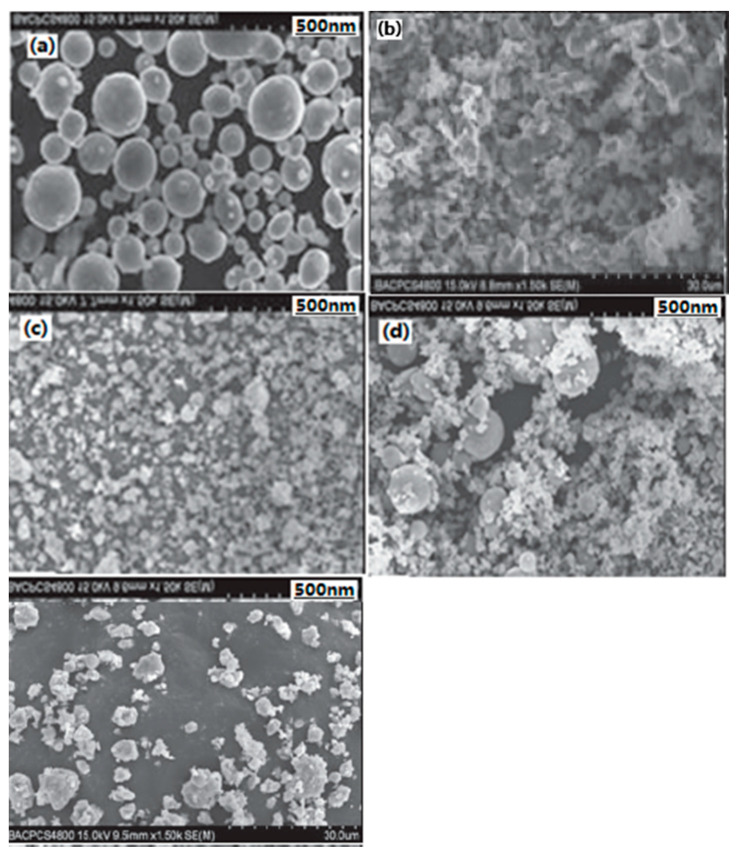
SEM photographs of different samples. Reproduced from [48], with permission from Gu Ti Huo Jian Ji Shu, 2014. (**a**) Raw Al; (**b**) raw B; (**c**) Al/B composite; (**d**) physical mixed; and (**e**) ball milling.

**Figure 10 nanomaterials-10-01039-f010:**
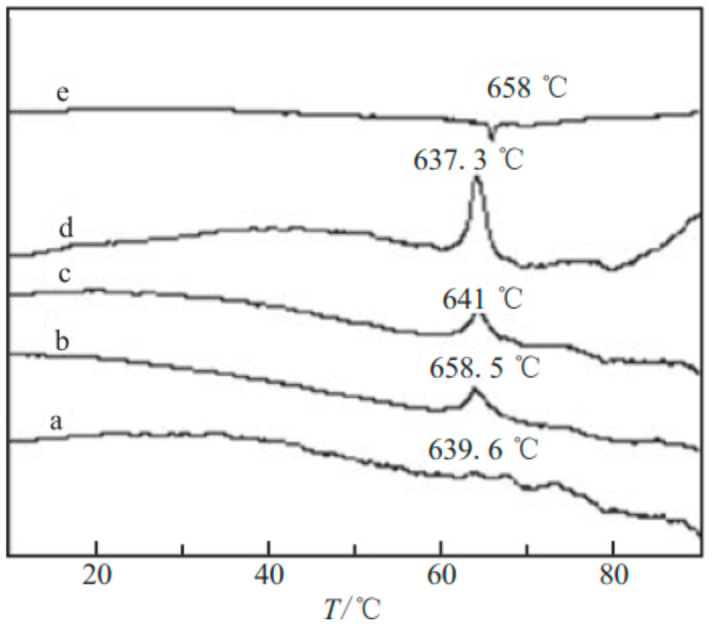
DSC of Al/B/Fe_2_O_3_ nano-sized composites. Reproduced from [48], with permission from Gu Ti Huo Jian Ji Shu, 2014.

**Figure 11 nanomaterials-10-01039-f011:**
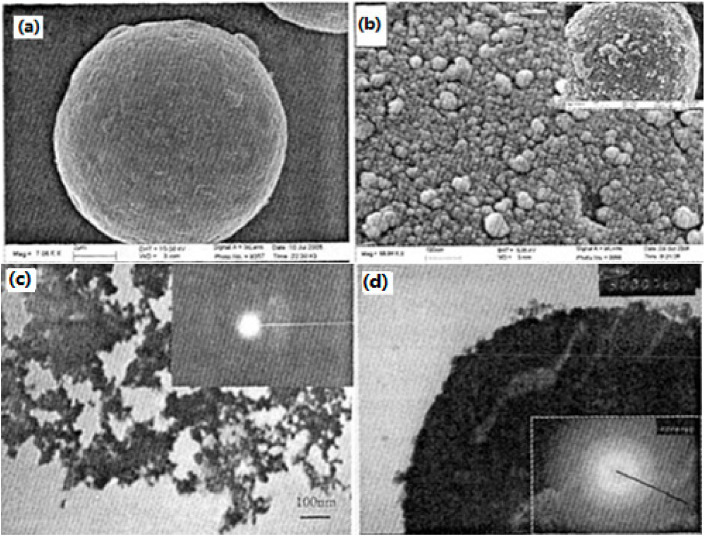
SEM and transmission electron microscopy (TEM) photographs of different samples. (**a**) Raw Al powder; (**b**) Al/B/Ni nano-sized composite particles. Reproduced from [36], with permission from Elsevier, 2016; (**c**) Ni/B nano-sized alloy; and (**d**) Al/B/Ni nano-sized composite particles. Reproduced from [50], with permission from Han Neng Cai Liao, 2009.

**Figure 12 nanomaterials-10-01039-f012:**
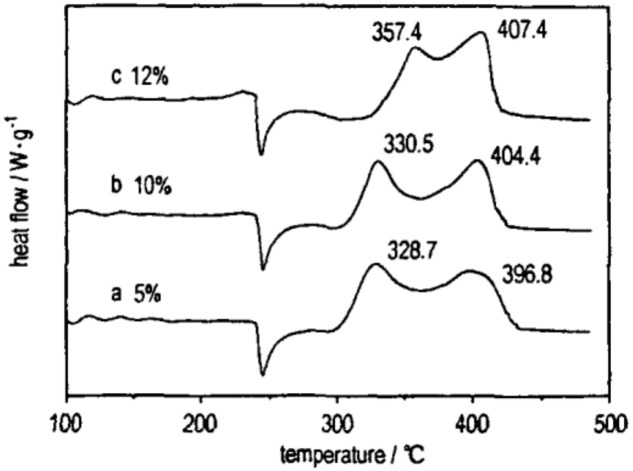
DTA curves of AP with different ratios of nano-sized Al/B/Ni composite particles. Reproduced from [50], with permission from Han Neng Cai Liao, 2009.

**Figure 13 nanomaterials-10-01039-f013:**
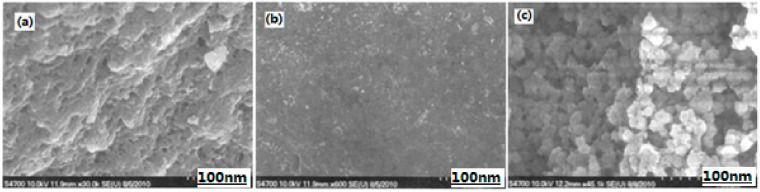
SEM photos of different particles. (**a**) Fe_2_O_3_ gas gel; (**b**) lowresolution microscopic of Al/RDX/Fe_2_O_3_ composite; and (**c**) high-resolution microscopic of Al/RDX/Fe_2_O_3_ composite. Reproduced from [51], with permission from Han Neng Cai Liao, 2011.

**Figure 14 nanomaterials-10-01039-f014:**
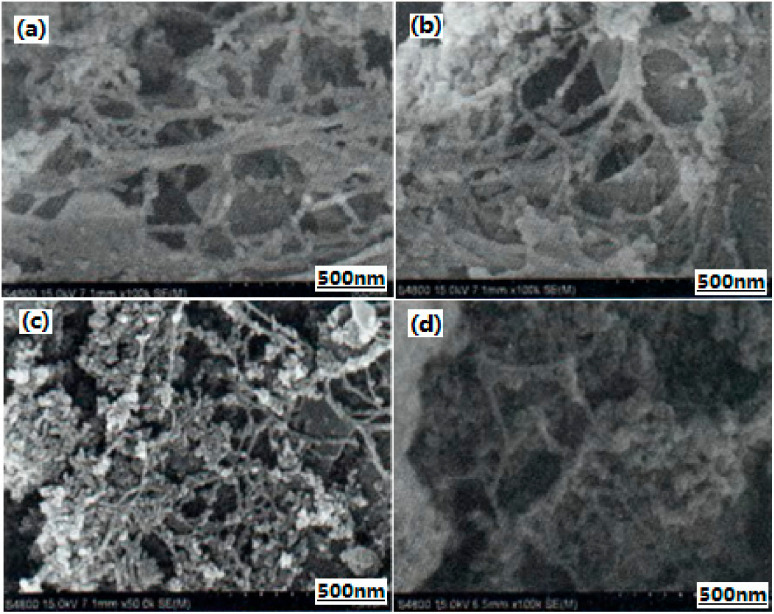
SEM photos of SiO_2_ xerogel and nano-sized Al/RDX/SiO_2_ composite particles. (**a**) SiO_2_ xerogel; (**b**) 30%; (**c**) 50%; (**d**) 70%. Reproduced from [53], with permission from Han Neng Cai Liao, 2017.

**Figure 15 nanomaterials-10-01039-f015:**
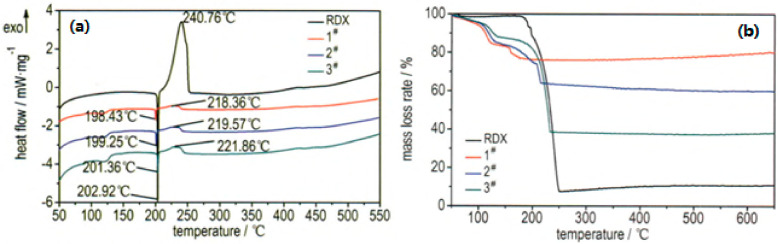
DSC (**a**) and TG (**b**) curves of pure RDX and nano-sized Al/RDX/SiO_2_ composite particles. 1# is 30%, 2# is 50%, and 3# is 70%. Reproduced from [53], with permission from Han Neng Cai Liao, 2017.

**Figure 16 nanomaterials-10-01039-f016:**
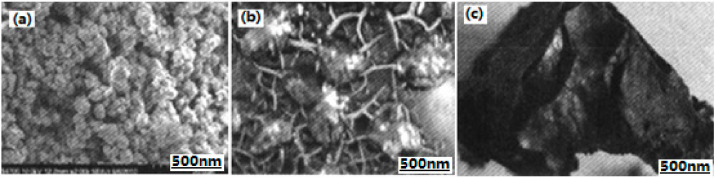
SEM photos of different particles. (**a**) Nano-sized Al/RDX/AP/SiO_2_ composite; (**b**) gel complex by vacuum freeze drying; and (**c**) gel complex by supercritical drying. Reproduced from [55], with permission from Nanjing University of Science and Technology, 2018.

**Table 1 nanomaterials-10-01039-t001:** Hazardous properties of RDX and different compositions.

Samples	Impact Sensitivity/cm	Standard Deviation	Friction Sensitivity/%	Detonation Velocity/m·s^−1^
RDX	22.5	0.04	96	6570
Al/RDX/Fe_2_O_3_	50.2	0.05	7	7185

Note: impact sensitivity test condition: sample mass is (35 ± 1) mg, drop weight is 2.5 kg, and relative huimidity is ≤80% at room temperature; friction sensitivity test condition: swing angle is (90 ± 1)°, gage pressure is 3.92 MPa, and sample mass is (20 ± 1) mg; detonation velocity test condition: sample shape is *Φ*10 mm × 10 mm and density is 1.55 g cm^−3^.

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
