# Peer review of "Al-Based Nano-Sized Composite Energetic Materials (Nano-CEMs): Preparation, Characterization, and Performance"

_nanomaterials, 2020, doi:10.3390/nano10061039_

Round 1

Reviewer 1 Report

Presented paper summarizes the available results on nanoenergetic systems, based on aluminum. The research in this are is highly important and the review seems to be interesting for community. However, in my opinion the authors miss some important research, e.g., from groups by Profs. Dreizin, Pivkina, Gromov. In their papers they explore the reactivity of nanometals (especially Al), present some developments in synthesis of nanocomposites (arrested milling, spray drying etc.), thermites, and its applications.

Author Response

Reviewer 1

Presented paper summarizes the available results on nanoenergetic systems, based on aluminum. The research in this area is highly important and the review seems to be interesting for community. However, in my opinion the authors miss some important research, e.g., from groups by Profs. Dreizin, Pivkina, Gromov. In their papers they explore the reactivity of nanometals (especially Al), present some developments in synthesis of nanocomposites (arrested milling, spray drying etc.), thermites, and its applications.

Reply: This paper is mainly focus on the Al-based composite, there is no nanometals in the text.

Nearly ten papers on Al-based nano-sized energetic materials from Prof. Dreizin, Pickina, Gromov, et al were referenced and cited in the text.

Reviewer 2 Report

In this submission to Nanomaterials, the authors present a review on the progress of Al-based nano-CEMs. In particular, the authors analyze the preparation methods and properties of Al-based nano-CEMs with an emphasis on the improved performances of Al-based nano-CEMs. The authors note that these structures are different from those of conventional micro-sized composite energetic materials (micro-CEMs), such as thermal decomposition and hazardous properties. The existing problems and challenges for the future work on Al-based nano-CEMs are discussed.

I consider this manuscript to be of interest to the readers of Nanomaterials, and I am supportive of publication with a minor note. In particular, there has been a previous review and work on the manufacture of aluminum nanocomposites and thermodynamic calculations to understand these effects:

https://doi.org/10.4028/www.scientific.net/MSF.678.1

https://pubs.acs.org/doi/full/10.1021/jp112258s

In particular, these prior works have reviewed and examined aluminum nanocomposites to understand how they can be leveraged for manufacturing and characterization, which should be mentioned as prior work in this area. With this minor revision, I would be supportive of publication in Nanomaterials.

Author Response

Reviewer 2

In this submission to Nanomaterials, the authors present a review on the progress of Al-based nano-CEMs. In particular, the authors analyze the preparation methods and properties of Al-based nano-CEMs with an emphasis on the improved performances of Al-based nano-CEMs. The authors note that these structures are different from those of conventional micro-sized composite energetic materials (micro-CEMs), such as thermal decomposition and hazardous properties. The existing problems and challenges for the future work on Al-based nano-CEMs are discussed.

I consider this manuscript to be of interest to the readers of Nanomaterials, and I am supportive of publication with a minor note. In particular, there has been a previous review and work on the manufacture of aluminum nanocomposites and thermodynamic calculations to understand these effects:

https://doi.org/10.4028/www.scientific.net/MSF.678.1

https://pubs.acs.org/doi/full/10.1021/jp112258s

In particular, these prior works have reviewed and examined aluminum nanocomposites to understand how they can be leveraged for manufacturing and characterization, which should be mentioned as prior work in this area. With this minor revision, I would be supportive of publication in Nanomaterials.

Reply: Thank you very much for your positive comments and references introduction. The manuscript was revised and improved according to the reference 1, while the reference 2 is on the relative stabilities of alane (AlH3) complexes with electron donors, it has little relation to this topic in my opinion, so it was not cited in the text.

Reviewer 3 Report

The manuscript reports the recent progress in preparation, performance, and characterization of Al-based nano-CEMs. But there are still some problems in this paper. Firstly, the paper is not well organized, most of the results are not well explained to be published as it is. Most of the results show SEM data, but do not describe any fundamental theory, properties, and application of nano-CHEMs in detail. Second, the part result and discussion is written too badly. We only know the experimental phenomena without deep scientific discussion. Third, the manuscript reads more like a technical report for the preparation of the materials without the figures and tables for characterization and performance of materials. 

The article in the current form was poorly written and it should be drastically improved. This manuscript is insufficient and incomplete to be published in this journal.

Author Response

Reviewer 3

The manuscript reports the recent progress in preparation, performance, and characterization of Al-based nano-CEMs. But there are still some problems in this paper. Firstly, the paper is not well organized, most of the results are not well explained to be published as it is. Most of the results show SEM data, but do not describe any fundamental theory, properties, and application of nano-CHEMs in detail. Second, the part result and discussion is written too badly. We only know the experimental phenomena without deep scientific discussion. Third, the manuscript reads more like a technical report for the preparation of the materials without the figures and tables for characterization and performance of materials.

The article in the current form was poorly written and it should be drastically improved. This manuscript is insufficient and incomplete to be published in this journal.

Reply: Thanks you for your comments, the improvements are as follows:

(1) Several fundamental theory and applications were added to the text.

(2) The English statements and scientific discussion were improved and supplemented in detail.

(3) Several characterization and application of materials were supplemented in the text.

(4) The references were typed according the regulation of the Journal.

Round 2

Reviewer 3 Report

This paper has been revised in many parts, but it is still insufficient to be published in this journal.